# Early Immunocastration of Pigs: From Farming to Meat Quality

**DOI:** 10.3390/ani11020298

**Published:** 2021-01-25

**Authors:** Daniela Werner, Lisa Baldinger, Ralf Bussemas, Sinje Büttner, Friedrich Weißmann, Marco Ciulu, Johanna Mörlein, Daniel Mörlein

**Affiliations:** 1Institute of Organic Farming, Johann Heinrich von Thuenen Institute, Trenthorst 32, 23847 Westerau, Germany; daniela.werner@thuenen.de (D.W.); lisa.baldinger@thuenen.de (L.B.); ralf.bussemas@thuenen.de (R.B.); friedrich.weissmann@gmx.de (F.W.); SinjeBuettner@gmx.net (S.B.); 2Department of Animal Sciences, University of Goettingen, Kellnerweg 6, 37077 Göttingen, Germany; marco.ciulu@uni-goettingen.de (M.C.); johanna.moerlein@uni-goettingen.de (J.M.)

**Keywords:** meat production, performance, boar taint, animal welfare, fatty acid composition

## Abstract

**Simple Summary:**

Immunocastration of boars to prevent boar taint in meat is usually performed during the fattening phase of pigs, which is sometimes impractical for pig fatteners. The aim of this study was to test the practicability of the immunization of suckling pigs against boar taint and to assess its influence on production performance and animal welfare in the fattening phase. The fattening and slaughtering performance as well as animal behavior and welfare did not differ between the standard and the earlier immunization. However, reliable avoidance of boar taint was not given for all animals when immunization was conducted very early at the piglet stage.

**Abstract:**

The study aimed to test a very early immunization of pigs to prevent boar taint with regard to its practicability and influence on production performance, its reliability in ensuring good meat and fat quality, and animal welfare. Immunization was already conducted at piglet production stage and could be easily integrated into routine vaccination (week 3) and weaning practices (week 7). The fattening and slaughter performance of the animals was not affected by the immunization regime and was within the usual range. In addition, there were no abnormalities in animal behavior and the prevalence of injuries caused by aggressive interactions. All animals were classified as infertile on the basis of the histological examination of the testicles. However, the testosterone levels measured at slaughter were significantly higher in animals of the early immunization regime than in animals subjected to the standard immunization regime. Androstenone and skatole levels as the main components of boar taint were, on average, higher and varied to a greater extent in early immunized animals. Furthermore, the comparison of the immunization scheme did not result in significant differences for meat quality and for fatty acid composition.

## 1. Introduction

Consumers increasingly criticize pork production systems in the context of animal welfare. Surgical castration without anesthesia, allowed in Germany until the end of 2021, is the subject of ongoing public and academic discussions. Alternatives to the surgical castration of pigs are the fattening of boars and immunocastration. 

Immunocastration for the purpose of controlling boar taint provides several public as well as agribusiness advantages over physical castration. Primarily, immunocastration improves animal welfare since it does not involve painful procedures and reduces antagonistic, aggressive behavior by entire male pigs [1,2]. When comparing the productive performance of barrows to immunocastrated pigs, feed efficiency and lean carcass yield are higher in immunocastrated animals [3], which entails economic and environmental benefits through reduced feed expenditures and a reduction of nitrogen emissions [4]. The active ingredient of the immunocastration is a protein that delays the onset of puberty by stimulating the pig’s natural immune system to produce antibodies that inhibit testes function. The process is based on an antigen–antibody reaction that is achieved via two injections of the gonadotropin-releasing hormone (GnRH)-analogon Improvac ^®^ (Zoetis, Berlin, Germany). Immunocastration requires two doses of the vaccine at least 4 weeks apart, with the second immunization usually administered 4 to 5 weeks before slaughter, which means that immunocastration is done in the fattening phase of pigs. This implies surplus work for pig fatteners compared to surgical castration of piglets, which is especially challenging when dealing with inhomogeneous pig groups, as is often the case in organic pig fattening systems. Immunocastration at an earlier age (the first immunization at 10 and the second at 14 weeks of age) than the recommended immunization scheme has been found to have a greater negative impact on testes structure, development, and function [5,6]. 

We have therefore tested the immunocastration in the phase of the piglet production on the basis of the assumption that immunization at an earlier stage might lead to a permanent impairment of testicular development. As the needed injections for immunocastration can be incorporated in the standard piglet health management schemes, such a change of time would open a new possibility for conventional and organic pig fattening systems in terms of reduced workload while benefiting from the higher biological productivity of immunocastrates compared to barrows. In organic pig production systems with longer suckling periods in particular, an early immunization scheme could be easily integrated in the operational workflow. Furthermore, an earlier immunization could reduce animal welfare problems due to a reduced sexual and aggressive behavior during the fattening phase.

The aim of the present study was therefore to test the practicability of the immunocastration at the piglet production stage, and its consequences on production performance, meat, and fat quality as well as animal behavior and welfare. To our knowledge, there are no studies with an implementation of immunocastration as early as in this study.

## 2. Materials and Methods

The trial was conducted at the experimental farm of Thuenen Institute of Organic Farming in Trenthorst, Germany, between June 2018 and November 2019. Animal husbandry followed the rules of European Commission Regulation 889/2008. The experiment was announced to the Schleswig Holstein Ministry of Energy Transition, Agriculture, Environment and Rural Areas on April 23, 2018 and acknowledged on May 8, 2018 (reference number V 244-23356/2018). 

### 2.1. Experimental Design and Animals

Data of 109 boars from 2 treatments was obtained during 3 consecutive trial runs with both treatments being present in every trial run (see Table A1 in Appendix A). A total of 55 boars were objected to an early immunization scheme (EARLY) and 54 to the standard (CONTROL) via 2 consecutive injections with the GnRH-Analogon Improvac ^®^ (Zoetis, Berlin, Germany). Treatment groups were kept pen-wise. Piglets originated from the institute’s herd of crossbred sows (German Landrace × Large White), which were inseminated with 7 individual Piétrain boars. Complete litters were assigned to 1 of the 2 treatments.

All animals of CONTROL groups received the first immunization on the starting day of the pre-fattening phase with an average group weight of 36.7 kg per animal (12 weeks). The second immunization was given 4 to 6 weeks prior to slaughter with an average group weight of 75.3 kg (19 weeks). The pre-fattening phase ended at an average group weight of 50 kg. The subsequent fattening phase ended at a live weight of 115 kg. Animals were classified according to their growth capacity (slow, standard, fast) to ensure the correct immunization dates for the second immunization. The standard class weight range was defined as the average weight at the start of the fattening phase ± 1 standard deviation. All animals of EARLY groups received the first immunization at 3 weeks of age (ø 6.3 kg) in combination with immunizations against mycoplasma and coli bacteria. The second immunization was conducted during weaning at 7 weeks of age (ø 16.4 kg). 

### 2.2. Housing and Feeding

Animals were housed in an experimental fattening unit consisting of 4 pens with a maximum stocking rate of 10 animals per pen. Pens had solid concrete floors with straw as litter material. Each pen had an indoor area of 1.5 m^2^ animal^−1^ including a temperature-controlled sleeping box and an automated feeding system on both sides of the pen. The adjacent outdoor run had an area of 1.2 m^2^ animal^−1^ and was equipped with drinking troughs and a roughage rack. 

Feeding was split in a pre-fattening and fattening phase with 2 associated compound feeds (pre-fattening: 88% dry matter, 19.2% crude protein, 13.2 MJ ME kg^−1^, 0.80 lysine/ME; fattening: 88% dry matter, 17.1% crude protein, 12.5 MJ ME kg^−1^, 0.74 lysine/ME). During the pre-fattening phase, feeding was semi ad libitum, and during the fattening phase, a weight-oriented feeding curve was applied.

Animals had free access to clover-grass silage (18.8% dry matter, 15.6% crude protein, and 13.4 MJ ME kg^−1^ per dry matter) with a daily allowance of 0.5 kg and 1.0 kg fresh matter animal^−1^ during the pre-fattening and fattening phase, respectively.

### 2.3. Fattening and Slaughtering Performance

The fattening performance was characterized by live weight development and feed efficiency. Individual live weights during the fattening phase were measured weekly to calculate live weight gains. The amounts of allocated feed and feed refusals were measured daily. Feed efficiency was calculated per group. 

Slaughter animals were transported pen wise at 5:40 a.m. to the nearby (13 km) small family abattoir. Animals were unloaded pen-wise and immediately slaughtered by the use of the electrical stunning method. The slaughtering performance included carcass characteristics and physical meat quality characteristics (Table A1), which were individually measured 24 h *post mortem* using the left half of the carcass. Dressing percentage was calculated from warm carcass weight and final body weight. Lean meat percentage was calculated via the Bonner Formula, using caliper measurements (SCAN-STAR K, Matthäus, Eckelsheim, Germany). Electrical conductivity was measured between the 13th/14th rib via LF-star pistol (Matthäus, Eckelsheim, Germany).

The whole data recording concerning fattening and slaughtering performance followed the German guidelines for official performance testing [7].

### 2.4. Boar Taint and Fatty Acid Composition

For the determination of androstenone and skatole levels, we extracted subcutaneous shoulder fat (SF) 1 day after the slaughter. For the determination of the fatty acid compositions of the intramuscular fat (IMF) and subcutaneous fat, we withdrew the loin muscle (*Musculus longissimus thoracis et lumborum (LTL))* at the 13th rib. Shoulder fat and LTL samples were vacuum-packed and stored at −20 °C until further analysis.

#### 2.4.1. Androstenone and Skatole 

Chemical analysis of androstenone and skatole was performed using stable isotope dilution analysis with headspace solid-phase microextraction followed by gas chromatography with mass spectrometry (SIDA–HS-SPME–GC/MS) using deuterium-labeled internal standards [8]; the protocol and equipment as described in [9] was used. Results are given as ng g^−1^ melted back fat.

#### 2.4.2. Determination of the Fatty Acid Composition in the Intramuscular Fat and Subcutaneous Fat

Sample preparation of adipose tissue was performed as described by Liu et al. [10]. Fatty acids of intramuscular fat were extracted from freeze-dried samples and derivatized according to the procedure by Du et al. [11] for triglycerides adopting a derivatization time of 1 h instead of 40 min. 

Fatty acid methyl esters were analyzed by gas chromatography with flame ionization detector (GC-FID). The gas chromatograph (GC; TRACE 1310) was equipped with an AS1310 autosampler; equipment was sourced from Thermo Fischer Scientific (Waltham, MA, USA). A Supelcowax−10 (30 m × 0.32 mm × 0.25 μm; Sigma-Aldrich Chemie GmbH, Munich, Germany) capillary column was employed for the separation. Oven temperature was held at 160 °C for 1 min, increased until 220 °C (heating rate: 10 °C/min), maintained at 220 °C for 3 min, increased again to 250 °C (heating rate: 10 °C/min), and as a last step was held at 250 °C for 3 min. Each sample was injected adopting a 1:50 split ratio, at 250 °C, using hydrogen as the carrier gas with a flow rate of 1.2 mL/min. The whole chromatographic run was completed in 16 min. The FID operated at 260 °C with an air flow of 350 mL/min, hydrogen flow of 35 mL/min, and makeup gas flow of 40 mL/min. Fatty acids were identified using the Supelco 37 Component FAME Mix (Sigma-Aldrich, Munich, Germany) and relative areas were analyzed using the Chromeleon Chromatography Data System (Version 7.2 SR9; Thermo Fischer Scientific, Waltham, MA, USA). All the analyses were performed in duplicate. 

### 2.5. Testicular Development and Immunization Status

Testes were weighed immediately after the slaughter. To determine the fertility status, we cut 1 cm³ cubes of testes and epididymis no later than 3 h after slaughtering. Subsequently the sample material was prepared with 4% formaldehyde solution for overnight transport to the Clinic for Cloven-hoofed Animals of the Faculty of Veterinary Medicine, University of Leipzig. Histological analyses of the samples were conducted the following day as described elsewhere [12]. Results were categorized into findings on testes (atrophication, hypoplasia) and epididymis (azoospermia, teratozoospermia, oligospermia).

To determine the immunization status, we collected blood samples during the exsanguination of the pigs. Blood serum samples were stored at −20 °C and analyzed at the end of the trial. Analyses of the testosterone level and the absolute antigen–antibody binding level in the blood serum were conducted at the Department of Behavioral Physiology of Livestock of the University of Hohenheim as described elsewhere [13].

### 2.6. Animal Behaviour and Welfare

Data on animal behavior was collected via direct observations at 2 dates for each trial group. Each observation lasted 3 h in the morning and 3 h in the afternoon. Two observers conducted the observation prior to the second immunization of the CONTROL group animals as an effect of the immunization is expected 2 weeks after the second immunization (the first observation at an overall mean age of 16 weeks and the second observation at an overall mean age of 18 weeks). Only negative behavior (pushing, biting, and mounting) was counted. Inter-observer reliability between the 2 observers was estimated by calculating prevalence-adjusted bias-adjusted Kappa (PABAK) values for all types of behavior [14]. The prevalence of scratches and penis injuries were documented as indicators of animal welfare. Scratches were documented during the fattening phase every 2 weeks. The left side of the body was divided into 8 parts, which were examined according to the Welfare Quality Assessment protocol for pigs [15]. Penis injuries were scored immediately after the slaughter, with score 0 = no bite marks, score 2 = small bite marks, score 3 = severe bite marks.

### 2.7. Statistical Analysis

All statistical analyses were performed using SAS 9.4. PROC MIXED and the following models were used for the analysis of fattening and slaughtering performance, meat quality and fatty acid composition, testes weight, and immunization status:(1)Live weight: y_ijk_ = µ + Imm_i_ + Trial run_j_ + Litter_k_ + ε_ijk_(2)Daily gain: y_ijkl_ = µ + Imm_i_ + Trial run_j_ + Litter_k_ + Start weight_l_ + ε_ijkl_(3)Feed efficiency: y_i_ = µ + Imm_i_ + ε_i_(4)Slaughtering performance: y_ijk_ = µ + Imm_i_ + Trial run_j_ + Litter_k_ + Slaughter weight_m_ + ε_ijkm_(5)Meat and fat quality: y_ijk_ = µ + Imm_i_ + Trial run_j_ + Litter_k_ + ε_ijk_(6)Testes weight: y_ijk_ = µ + Imm_i_ + Trial run_j_ + Litter_k_ + Age at slaughter_n_ + Slaughter weight_m_ + ε_ijknm_(7)Immunization status: y_ijk_ = µ + Imm_i_ + Trial run_j_ + Litter_k_ + Age at slaughter_n_ + ε_ijk_

To analyze the effects of the different immunization schemes on testicular histology and animal welfare indicators, we compared the frequencies of categories and scores using PROC GLIMMIX (chi^2^ test, binomial distribution) and the following models: (8)Histology characteristics: y_ijk_ = µ + Imm_i_ + Trial run_j_ + ε_ij_(9)Behavior: y_ijk_ = µ + Imm_i_ + Trial run_j_ + Day section_o_ + ε_ijo_(10)Appearance of scratches: y_i_ = µ + Imm_i_ + ε_i_
where µ = overall mean, Imm_i_ = fixed effect of immunization scheme (i = CONTROL, EARLY), Trial run_j_ = fixed effect of trial run (j = 1, 2, 3), Litter_k_ = random effect of litter (k = individual litter), Start weight_l_ = continuous linear effect of live weight (l = individual live weight), Slaughter weight_m_ = continuous linear effect of slaughter weight (m = individual slaughter weight), Age at slaughter_n_ = continuous linear effect of age at slaughter (*n* = individual age at slaughter), Day section_o_ = fixed effect of time of observation (o = late morning, afternoon), ε = residual error.

Except for feed efficiency, the individual animal was the experimental unit. Feed efficiency was analyzed per pen. Litter number was recorded to include the effect of the sow on production performance.

Probability values < 0.05 were interpreted as indicating statistically significant differences. 

Due to the low sample size, penis injuries were evaluated descriptively.

## 3. Results

### 3.1. Fattening and Slaughtering Performance

The fattening phase in this trial comprised a live weight range from 25.3 to 51.1 kg (CONTROL) and 23.4 to 50.7 kg (EARLY) in the pre-fattening phase and ended mean live weights of 115.2 (CONTROL) and 114.1 (EARLY), respectively. No significant differences were found for live weight development between the trial groups. Feed conversion did not differ significantly between the trial groups during the two fattening phases, but it did differ between trial runs in the fattening phase. The daily weight gain of the EARLY pigs in the pre-fatting phase was significantly higher by 41 g than that of the CONTROL pigs. Daily weight gain from start to slaughter did not differ significantly between trial groups (903 and 916 g for CONTROL and EARLY pigs, respectively).

No significant differences in the slaughtering performance were found between the trial groups except for electrical conductivity 24 h postmortem, which was higher in the EARLY pigs. 

For detailed results, see Table A2.

### 3.2. Boar Taint and Fatty Acid Composition

Significant interactions between the immunization scheme and trial run were found for androstenone and skatole (*p* = 0.032 and 0.001, respectively). Androstenone concentrations in the EARLY pigs in the third trial run were lower compared to trial runs one and two and did not differ from values obtained in the CONTROL pigs in all trial runs. Skatole concentrations in the EARLY pigs in the first trial run were the highest among all trial groups. The other trial groups did not differ from each other (Table A3) with respect to skatole concentrations. The distribution of individual androstenone and skatole concentrations showed a wider dispersion within the EARLY group in comparison with the CONTROL group (Figure 1). On the basis of cut off values of 250 ng of skatole or 1.000 ng of androstenone per grams of fat as boundaries for boar tainted meat, we found that 7% (*n* = 4) of the EARLY pigs would have been classified as potentially tainted.

The percentage of monounsaturated (MUFA) and polyunsaturated (PUFA) fatty acids in the SF and in the IMF differed between the trial groups (Figure 2). While the fat of the CONTROL pigs had a significantly higher content of MUFA than EARLY pigs, it was the other way round for PUFA content (*p* < 0.001 and *p* = 0.015, respectively). The content of saturated fatty acids (SFA) did not differ significantly between the immunization schemes. Generally, SF had a higher content of PUFA and a lower content MUFA than IMF.

### 3.3. Testicular Development and Immunization Status

Testes weight of the EARLY pigs was significantly higher and showed a higher variation than that of the CONTROL pigs. The immunization scheme significantly influenced testicular development. While the testes of the EARLY group pigs were mainly underdeveloped (hypoplasia), the testes of the CONTROL pigs mainly showed signs of tissue shrinkage (atrophia) (see Table 1). Additionally, atrophication of the testes of the EARLY pigs was less advanced than in the CONTROL pigs (see Figure 3).

Histological results showed that none of the pigs in either trial groups were reproductive. While the number and quality of sperm in the CONTROL pigs was lower (oligo- and, teratozoospemia), the percentage of pigs that developed no sperm at all (azoospemia) was higher in EARLY group pigs (see Table 1).

Testosterone levels and GnRH binding differed significantly between the trial groups and a significant interaction of immunization scheme and trial run was found in the case of testosterone. Age at slaughter had no significant effect on testosterone values, whereas GnRH binding decreased with increasing age at slaughter. Regardless of the lower testosterone levels of EARLY pigs in the third trial run compared to the first and second run (4.58 vs. 17.80 and 11.60 ng/mL), the levels still were markedly higher than for CONTROL pigs (0.29, 0.50, and 0.21 ng/mL in the first, second, and third run respectively). Testosterone levels in the serum were measured with a mean time span (min,max) between the second immunization and slaughter of 6 (4,9) and 17 (12,23) weeks for the CONTROL and EARLY pigs, respectively, and they showed higher variation in EARLY pigs (Figure 4). 

GnRH binding was significantly higher in the CONTROL pigs by 16.5% and was influenced by the age at slaughter (see Table A4), whereas GnRH binding decreased with increasing age. Although GnRH binding of nearly 46% (*n* = 21) of the EARLY pigs was at the same level measured for the CONTROL pigs (>30%), the testosterone levels of the EARLY pigs showed highly variable values compared to the CONTROL group (see Figure 5). GnRH binding showed great variation in the EARLY pigs and was negatively correlated to the testosterone values in the serum (r = −0.583; *p* < 0.001; *n* = 45).

### 3.4. Animal Behaviour and Welfare

The immunization scheme had no significant effect on the occurrence of antagonistic behavior (*p* = 0.8955), although the counts for biting, pushing, and riding were slightly higher for the EARLY pigs (Figure 6).

This corresponds to the results showing the relative frequencies of injuries, with the EARLY pigs showing a slightly but not significantly higher percentage of injuries in total (see Table 2). The penis rating scores for *n* = 2 CONTROL and *n* = 17 EARLY pigs could not be deducted as the penis could not be removed from the shaft. However, the majority of penis tips was intact (92.3% and 92.1% for the CONTROL and EARLY pigs, respectively).

## 4. Discussion

The aim of this study was to test the practicability of an immunization of suckling pigs against boar taint and its influence on production performance and animal welfare in the fattening phase.

During the trial phase, the early immunization scheme was easily integrated in the standard management procedures of the organic piglet production system of the research station and did not create an additional workload. The first immunization of the piglets at three weeks of age was combined with the standard immunization scheme. The immunization at seven weeks of age could be administered during the separation process of the piglets during weaning. When using the same immunization scheme in conventional systems, the first immunization could be administered during weaning, while the second immunization would have to be administered during the rearing phase. Piglets tolerated both immunizations well, and in the subsequent fattening, no additional treatments were necessary.

Daily weight gain during the complete fattening period did not differ significantly between the trial groups and was 903 and 916 g for the CONTROL and EARLY pigs, respectively. This is in accordance to the values found for immunocastrated organic pigs using the standard immunization scheme [16] but lower than the values for conventionally kept pigs with an early immunization scheme (8 and 12 weeks of age) [17]. Although feed intake during the pre-fattening phase did not differ significantly between the trial groups, it was higher in the EARLY pigs. This led to a significantly higher daily gain of the EARLY pigs in the pre-fattening phase, which can be explained by the higher growth response of the pigs to the second immunization. After the first immunization, pigs physiologically resemble entire male pigs, whereas after the second immunization, the metabolism changes and feed intake increases whereas feed conversion decreases [18]. Carcass yield and lean meat did not differ between trial groups and are similar to values found for immunocastrated pigs, [16,19,20]. Electrical conductivity was significantly higher for the EARLY pigs; nevertheless, this did not indicate inferior meat quality in terms of pale, soft, and exudative (PSE) meat as the time-dependent threshold value of 6 mS/cm 24 h postmortem was not exceeded [21,22].

The percentage of underdeveloped testes was higher in early immunized pigs, but testes weights of this group showed a greater variation compared to the CONTROL pigs. This is in accordance with Sauer et al. [12], who also found a greater variation for testes weights in pigs immunized at an age of 11 weeks and 18 or 21 weeks. In some of the EARLY pigs, an only slightly or moderately less advanced atrophication of testes was associated with testes weights that were comparable to values for entire male pigs. They were markedly higher than testes weights of pigs in the early immunization schemes of Sladek et al. [23], which received a first dose at 8 weeks and a second dose 8 or 15 weeks prior to slaughter. The individual time span between the last immunization and slaughter in this study was 12 to 23 weeks for the EARLY group pigs and a resumption of steroidogenesis, which is accompanied by a rise of testosterone values, might be an explanation for higher testes weights [17]. Highly individual and variable time spans of 10 to 24 weeks from the last immunization to a resumption of testes function have been described earlier [24,25]. In these studies, the determined testosterone threshold in the serum for an assumed resumption of the testicular function was 0.5 ng/mL and therefore distinctly lower than values found for the EARLY group pigs in this study. However, in accordance to the findings of Einarsson et al. [6], the incidence of azoospermia was higher in EARLY immunized pigs, implicating a permanently affected spermatogenesis for 50% of the EARLY pigs.

Highly variable GnRH-binding percentages, which were negatively correlated to testosterone for the EARLY pigs, indicate an onset of testosterone production for some EARLY group pigs. This might be due to a diminishing effect of GnRH immunization during the time from the second immunization to slaughter. Testosterone values higher than 20 ng/mL in the serum were found for animals that were slaughtered 17 weeks after the second immunization and later. This is in contrast to Zamaratskaia et al. [26], who found GnRH antibody titers 22 weeks after the second immunization that were not accompanied by a marked increase of testosterone in the plasma. Therefore, they suggested that it might take more than four months for immunocastrated pigs to regain their reproductive potential. Due to the high variability found for testosterone values and GnRH binding in combination with morphological and histological testes characteristics a resumption of testes function before slaughtering for some of the EARLY pigs can be assumed.

A significant interaction between trial run and immunization scheme on testosterone, androstenone, and skatole values was found in this study, with levels being highest for the EARLY pigs in trial run one and lowest in trial run three. The formation of skatole and androstenone in entire male pigs are interdependent and controlled by the production of anabolic hormones [27]. As already described above, it is possible that for the EARLY pigs in trial run one the onset of testosterone production stimulated the formation of androstenone and skatole to a higher extent than in trial run two and three as age at slaughter and time span between second immunization and slaughter were higher for these pigs. Mean age at slaughter was six days higher in pigs of trial run one and the time between second immunization and slaughter was one week longer than for pigs of trial run three (18 vs. 17 weeks, respectively). Furthermore, the slaughter dates in trial run one were distributed over a longer period than in trial runs two and three. With increasing experience regarding the immunization of piglets against boar taint in the course of the project, the uniformity of procedures was improved over the three consecutive trial runs. This might be an explanation for the higher variation in time spans in trial run one and the resulting variations in testosterone, androstenone, and skatole. Studies on the influence of age on the contents of androstenone and skatole in entire male pigs reveal conflicting results. An age-related rise of androstenone was described for entire male pigs by Bonneau et al. [28]. The authors found an effect of age on androstenone levels in young, light boars, whereas in older boars the influence of body weight was more pronounced. This is in contrast to Thomsen et al. [29], who found no influence of age on androstenone levels in boars. Zamaratskaia et al. [30] described an age-related rise in skatole values in pigs from 180 d upwards. However, age at slaughter had no statistically significant effect on androstenone and skatole values in our study. There are also studies that show an influence of season on pubertal maturation and boar taint compounds, i.e., increasing pubertal activity and/or androstenone contents in the fat of entire male pigs with decreasing length of natural light exposure (see, for example, [29]). Animals of the trial runs one, two, and three were fattened in October/November, March/April, and July/August, respectively, and androstenone values decreased from trial run one to three. However, the pigs in this study would not be classified as entire male pigs on the basis of their immunization status. As described above, the time span between the first and the second immunization was the longest in trial run one and the slaughter dates distributed over a longer period than in trial runs two and three. Therefore, an influence of slightly skewed immunization dates between the trial runs on androstenone values seems more likely. Hence, a combination of a higher age at slaughter with a decreased effect of GnRH immunization during the time from the second immunization to slaughter on testosterone production and therefore on the formation of androstenone and skatole is likely.

For 7% of the EARLY pigs, threshold values of 250 ng skatole and 1000 ng androstenone per grams of melted shoulder fat were exceeded. Furthermore, the variability of measured boar taint compound values was higher than for the CONTROL pigs. None of the CONTROL pigs would have been classified as boar tainted. While production of sausage with tainted meat is possible depending on processing technique and blending percentage [31,32], the regular immunization scheme could be favorable for fresh meat production as the early immunization scheme used in this study could not prevent the possibility of boar tainted meat as reliable in all animals.

The contents of fatty acids found in this study are within the ranges found by other authors for immunocastrated pigs [16,17,33]. The percentage of MUFA and PUFA in the SF and IMF differed significantly between trial groups, while SFA percentage did not differ significantly. This is in contrast to Zoels et al. [17], who found no differences for MUFA and PUFA in pigs immunized at 8 and 12, or 12 and 16 weeks, respectively. Nevertheless, the differences between immunization schemes are numerically small and of little practical importance for commercial fresh meat production. However, percentage of PUFA in SF exceeded the recommendations of a maximum content of 12 to 15% [34,35], risking cutbacks in product quality for processed products (e.g., dry fermented sausages, cured ham). The feeding of silage is known to have an effect on the level of PUFA in the intramuscular fat of pigs [36,37]. The considerable PUFA levels observed can be attributable to the inclusion of silage in the diet.

Observer alignment for the survey of animal behavior resulted in a PABAK value of 0.88, which shows high a conformity of observations according to Gunnarsson et al. [38]. Problematic animal behavior did not differ significantly between trial groups but was higher for EARLY pigs. It has been unequivocally proven that immunocastration reduces aggressive and sexual behavior in pigs [2,19,39]. Andersson et al. [19] found significantly higher negative animal behavior in pigs subjected to the standard immunization scheme prior to the second immunization when compared to pigs early and fully immunized at 10 and 14 weeks, respectively. However, these differences were eradicated after the second immunization of the standard immunization group. Animal behavior in our study was observed prior or shortly after the booster immunization of the CONTROL, and a reducing effect of the second immunization of the CONTROL pigs on problematic behavior is likely. It is possible that this effect was already diminished in the EARLY pigs, as behavioral observations were conducted 9 to 11 weeks after the second immunization of this group. Nevertheless, the frequency and distribution of skin lesions did not differ significantly between the immunization groups throughout the fattening period, indicating no increased aggressive behavior of the EARLY pigs during the whole production period. Sexually motivated behavior such as riding and/or mounting did not differ between groups, and the percentage of penile injuries did not differ between the trial groups but the proportion of penis tips that could not be removed from the shaft after slaughter was higher in the EARLY pigs (31% vs. 4%). This indicates an arrested puerperal development in the EARLY pigs as the penile frenulum, which prevents the extrusion of the penis, was not disrupted, which is usually the case during puberty [40].

## 5. Conclusions

An early immunization scheme could be integrated into the usual work routines of vaccination (week 3) and weaning (week 7) of organic piglet production without negative consequences on production performance compared to the standard immunization scheme. Although testosterone values were higher in early immunocastrated pigs, negative animal behavior did not increase, and on the basis of the histological examination of the testicles, we found that all animals could be classified as infertile. Even though the time span between second immunization and slaughter varied between 13 and 23 weeks and led to androstenone and skatole levels that showed both higher average values and a significantly greater variation in early immunized animals, the percentage of boar-tainted animals in this group was only 7%.

We therefore conclude that the implementation of an early immunization scheme at piglet production stage is possible. Yet, the dose applied in our trial resulted in a higher frequency boar taint than in the standard immunization scheme.

## Figures and Tables

**Figure 1 animals-11-00298-f001:**
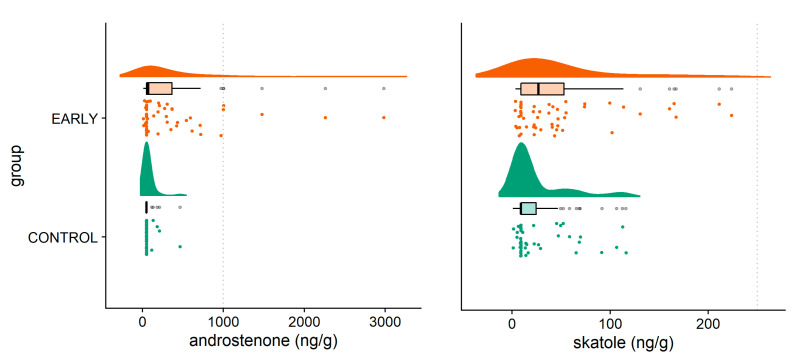
Distribution of androstenone (**left**) and skatole (**right**) concentrations as measured in the shoulder fat (SF) according to the immunization scheme.

**Figure 2 animals-11-00298-f002:**
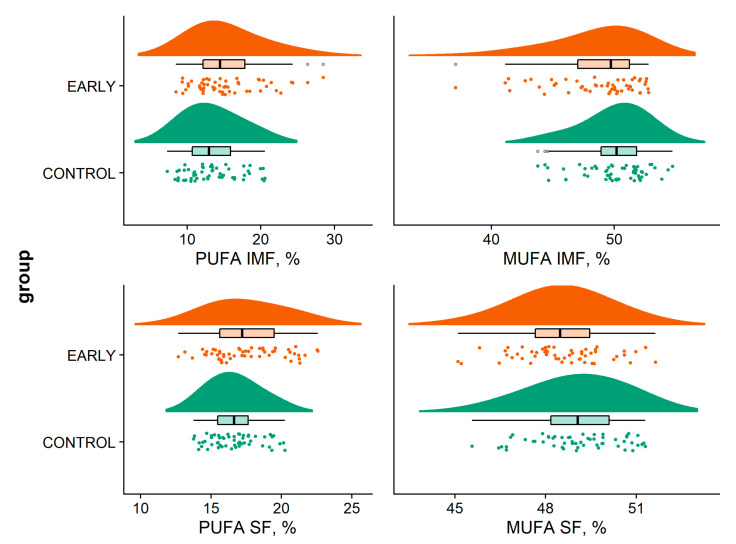
Fatty acid composition (sum of monounsaturated fatty acid (MUFA) and polyunsaturated fatty acid (PUFA), respectively) in subcutaneous fat (SF) and intramuscular fat (IMF) according to immunization scheme.

**Figure 3 animals-11-00298-f003:**
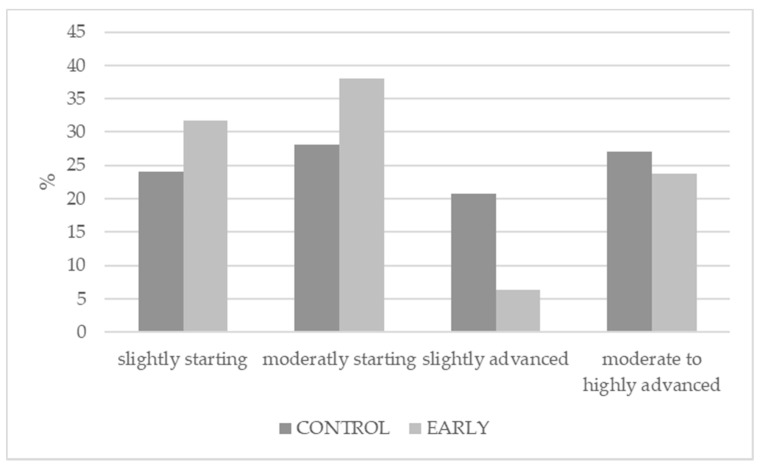
Atrophication classification according to immunization scheme.

**Figure 4 animals-11-00298-f004:**
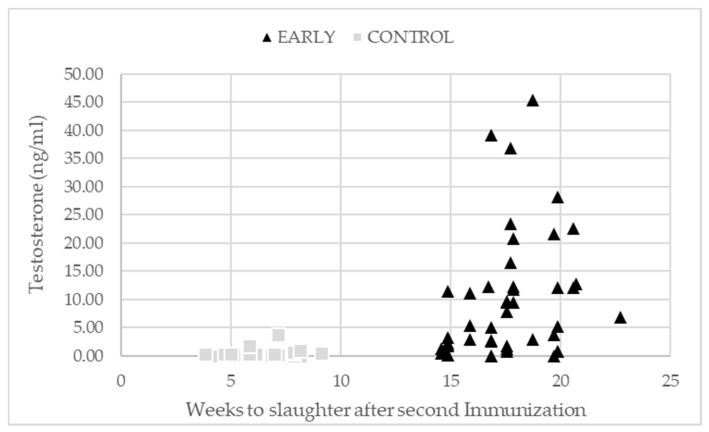
Testosterone levels depending on the number of weeks from the second immunization to slaughter according to the immunization scheme.

**Figure 5 animals-11-00298-f005:**
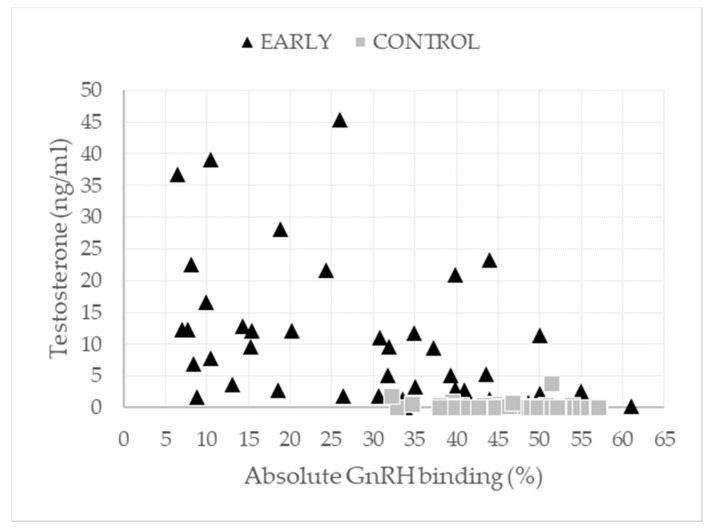
Testosterone levels and gonadotropin-releasing hormone (GnRH) binding according to the immunization scheme.

**Figure 6 animals-11-00298-f006:**
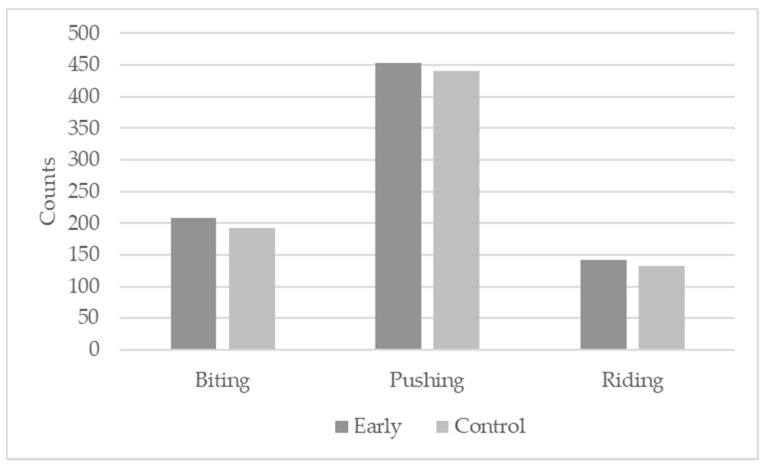
Counts of antagonistic behavior according to the immunization scheme.

**Table 1 animals-11-00298-t001:** Testes weight and development according to immunization scheme.

	Immunization		*p*-Values			
	Control	Early	Imm	Trial Run	Age at Slaughter	Slaughter Weight
LS Means	*n* = 54	*n* = 55				
Testes weight (g)	225	345	*<0.001*	*0.384*	*<0.001*	*0.013*
*Standard error*	*23.61–34.55*					
Relative frequencies (%)	*n* = 106	*n* = 108				
*Testes*						
Hypoplasia	9.4	41.7	*<0.001*	*0.144*	*-*	-
Atrophied	90.6	58.3	*<0.001*	*0.144*	*-*	-
*Epididymis*						
Azoospermia	20.8	51.9	*<0.001*	*0.120*	*-*	-
Oligospermia	75.5	41.7	*<0.001*	*0.649*	*-*	-
Teratozoospermia	78.3	46.3	*<0.001*	*0.171*	*-*	-

Immunization scheme (Imm).

**Table 2 animals-11-00298-t002:** Relative frequencies of injuries of different body parts according to the immunization scheme and the trial run.

	Imm		Trial Run			*p*-Values	
Relative Frequencies (%) of Injuries	Control	Early	1	2	3	Imm	Trial Run
	*n* = 266	*n* = 259	*n* = 200	*n* = 164	*n* = 161		
Total	41.7	45.6	58.0	40.9	28.6	*0.625*	*<0.001*
Ears	15.8	17.4	18.5	20.1	19.5	*0.722*	*0.051*
Head to shoulder	24.1	23.2	32.5	20.1	16.2	*0.602*	*<0.001*
Shoulder to flank	14.7	19.3	24.0	14.6	10.6	*0.238*	*0.004*
Ham	6.0	5.8	11.0	4.3	1.2	*0.685*	*0.002*
Legs	1.1	2.3	3.5	1.2	0.0	*0.393*	*0.431*
Foreskin	0.4	1.5	0.0	0.6	2.5	*0.157*	*0.418*
Testes	0.7	0.0	0.0	1.2	0.0	*0.960*	*0.998*
Tail	2.6	3.9	4.5	0.6	4.4	*0.439*	*0.150*

Immunization scheme (Imm).

## Data Availability

The data presented in this study are available on request from the corresponding author.

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
