# Peer review of "Early Immunocastration of Pigs: From Farming to Meat Quality"

_animals, 2021, doi:10.3390/ani11020298_

Round 1

Reviewer 1 Report

Surgical castration is commonly used to deal with the problem of boar taint on meat quality, especially in developing countries, but animal welfare has prompted the occurrence of a new approach to solve the problem of boar taint in pig industry. Immunocastration of boars is a good way to prevent boar taint in meat quality in a condition of animal welfare. Traditionally, immunocastration was conducted at least 4 weeks and 4 to 5 weeks before slaughter, which leads to surplus work for pig fatteners compared to surgical castration of piglets. This study explored the effect of earlier immunocastration on production performance and demonstrated that its practicability. The biggest advantage of this kind of earlier immunocastration is that it integrated into routine vaccination and did not increase surplus work in pig production. The design of experiment and language are excellent, statistical methods are appropriate, and results are reliable, which provide valuable information for the application of earlier immunocastration in pig industry in a condition of animal welfare. It can be accepted after minor revision.

Some minor errors in the format. For example:

  1. Line 206, the bracket was lacked after the word of “EARLY”.
  2. Line 262, Table 1 was not cited, and “Relative Frequencies (%) of …” is not completed and the same error was happened in Table 2.
  3. Line 238, The dot is redundant in the “(Figure 1.)” .
  4. Line 242, 271, 284, 295, uniforms the format.
  5. The format of “P value” should be uniform in the whole manuscript.
  6. line 259, 261, correct the errors “(see Error! Reference source not found..)”.

Author Response

Dear reviewer, thank you for your time and the positive evaluation. PLease find our detailed comments attached.

Reviewer 2 Report

Interesting and good paper

Material and methods:
some details that should be specified:
it would be enlightening to make a table indicating the number
of animals in each trial and for each treatment.
Also the number of
pens. Are both treatments in the same pen?

Differences in boar taint compounds can be due to the season effect?
Can the differences between trials, on sexual odor components, be explained
to the effect of the season in which the animals were fattened?

Statistical analysis
Complete some details or clarify

What is the experimental unit?
The effect of the litter is not explained before. Clarify. Are there brothers in different pens?
Have you considered the pen effect?
Have you tak
6. bTestes weight.... a -+- is missing between age and slaughter weight

Author Response

(The authors gave the same response as above.)

Reviewer 3 Report

Contents of all sections are appropriate and adequate. Descriptions were made very carefully and detailed. Generally manuscript is written duly and correctly with quality English language.

I wonder if it is an explanation of variation between consecutive trails. Considering progress in every trail is it possible that persons who made vaccination were more experienced from trial to trail and this can impact the results?

Author Response

(The authors gave the same response as above.)
